# The Roles and Associated Mechanisms of Adipokines in Development of Metabolic Syndrome

**DOI:** 10.3390/molecules27020334

**Published:** 2022-01-06

**Authors:** Ji-Eun Kim, Jin-Sun Kim, Min-Jee Jo, Eunjung Cho, Shin-Young Ahn, Young-Joo Kwon, Gang-Jee Ko

**Affiliations:** 1Department of Internal Medicine, Korea University Guro Hospital, Seoul 08308, Korea; beeswaxag@naver.com (J.-E.K.); dalky35@naver.com (J.-S.K.); minjeeyoyo@naver.com (M.-J.J.); icdej@naver.com (E.C.); sypooh712@naver.com (S.-Y.A.); yjkwon@korea.ac.kr (Y.-J.K.); 2Department of Internal Medicine, Korea University College of Medicine, Seoul 02841, Korea

**Keywords:** adipokine, metabolic syndrome, glucose intolerance, lipid metabolism

## Abstract

Metabolic syndrome is a cluster of metabolic indicators that increase the risk of diabetes and cardiovascular diseases. Visceral obesity and factors derived from altered adipose tissue, adipokines, play critical roles in the development of metabolic syndrome. Although the adipokines leptin and adiponectin improve insulin sensitivity, others contribute to the development of glucose intolerance, including visfatin, fetuin-A, resistin, and plasminogen activator inhibitor-1 (PAI-1). Leptin and adiponectin increase fatty acid oxidation, prevent foam cell formation, and improve lipid metabolism, while visfatin, fetuin-A, PAI-1, and resistin have pro-atherogenic properties. In this review, we briefly summarize the role of various adipokines in the development of metabolic syndrome, focusing on glucose homeostasis and lipid metabolism.

## 1. Introduction

With the increasing prevalence of overweight and sedentary lifestyles worldwide, a global epidemic of metabolic syndrome has begun. According to a 2015 global survey of obesity, the prevalence of obesity has doubled in 73 countries since 1980 [1]. The prevalence of metabolic syndrome varies based on the criteria defined by various organizations, but metabolic syndrome is estimated to occur in approximately one-quarter of the world population [2].

Metabolic syndrome is a combination of interrelated conditions that often occur together, including obesity, insulin resistance, glucose intolerance, hypertension, and dyslipidemia [3]. Metabolic syndrome is diagnosed as the presence of at least three of the following five characteristics: high waist–hip ratio, high blood pressure, elevated blood sugar level, increased triglycerides (TGs), and low high-density lipoprotein (HDL) cholesterol [4]. Metabolic syndrome is important because of its association with an increasing prevalence of diabetes and a higher risk of cardiovascular events such as heart disease and stroke, which have become major public health issues [5]. Dysregulation of certain adipokines can promote pathogenic conditions associated with obesity, lipid accumulation, and insulin resistance. These increase the risk of atherosclerosis [6]. Here, we summarize the molecular mediators that are thought to be associated with metabolic syndrome and can serve as potential therapeutic targets to prevent or treat this syndrome.

## 2. Adipose Tissue as a Critical Endocrine Organ Causing Metabolic Syndrome

Central obesity is the main cause of the etiological cascade of metabolic syndrome. Abnormal fat distribution, rather than adiposity itself, is a more important risk factor for obesity-related disorders [7,8]. Recent research has indicated that visceral adipose tissue is ‘ectopic fat’ that originates from subcutaneous adipose tissue as overflow fat beyond the storage capacity for extra energy [9,10]. As ‘ectopic fat’, visceral adipose tissue is associated with insulin resistance, lipoprotein metabolism, and elevated blood pressure [11]. Adipose tissue is an endocrine organ that expresses and secretes various adipokines [12]. Adipose tissue includes adipocytes, pre-adipocytes, adipose tissue macrophages, other immune cells, and vascular components. Several factors that are mainly secreted from adipocytes and adipose tissue macrophages contribute to the development of metabolic syndrome [13,14]. More recently, the chronic inflammatory condition that accompanies central obesity has been implicated as a major factor in the development of both metabolic syndrome and its associated pathophysiological consequences [15]. In this context, the role of factors derived from adipose tissue in the etiology of metabolic syndrome and the hypothesized associated mechanisms are discussed in Table 1.

For this review, a literature search was performed using electronic databases such as PubMed and Embase. The search strategy included any original article or review on the relationship between metabolic syndrome and individual adipokines. We used combinations of the following search terms: metabolic syndrome, obesity, adipokine, cytokine, leptin, adiponectin, visfatin, fetuin-A, resistin, omentin-1, lipocalin-2, asprosin, and neuregulin 4. In vivo or in vitro experimental studies for revealing associated mechanisms and related clinical studies were reviewed.

## 3. Dysregulation of Adipokines in Metabolic Syndrome

### 3.1. Leptin

Leptin is a 16 kDa cytokine primarily produced by white adipose tissue; its secretion increases according to the volume of adipose tissue or adipocytes [53].

#### 3.1.1. Experimental Studies

Leptin activates the Janus kinase (JAK)/signal transducer and activator of transcription (STAT3) proteins in the hypothalamus, thereby reducing orexigenic peptides such as neuropeptide Y and agouti-related protein and curtailing hunger and food intake [54]. Leptin improves insulin resistance through its metabolic and secondary effects following weight loss. In *ob/ob* mice, glucose and insulin levels were decreased within a few hours after leptin replacement despite the absence of a change in the mice’s weights [55], whereas administration of leptin antagonists increased the levels of blood glucose and insulin prior to changes in body weight [56]. Intravenous leptin injection normalized hyperglycemia and hyperinsulinemia and increased insulin sensitivity in lean mice [19]. Furthermore, leptin acts as an ‘adipostat’ to maintain the number of adipocytes [57]. Direct exposure to high leptin concentration suppresses the proliferation of pre-adipocytes and vascular stromal cells [17]. However, leptin signaling for activation of mitogen-activated protein kinase (MAPK) and STAT pathways has pro-adipogenic actions on subcutaneous pre-adipocytes, leading to hyperplasia and hypertrophy of adipocytes in vitro [16]. Leptin treatment was found to induce the expression of factors associated with adipogenesis and lipogenesis, including PPARγ and proinflammatory cytokines such as TNF-α, IL-10, and IL-6 [57]. Leptin also enhanced the formation of lipid droplets in pre-adipocytes in an mTOR-signaling-dependent manner [58]. Based on these intracellular signaling pathways, leptin induces pre-adipocyte differentiation even in the absence of insulin [58], which suggests a role for leptin in restoring local adipogenesis independent of insulin. Leptin plays a role in lipolysis, the process of mobilizing stored energy in the form of triglyceride in white adipose tissue. Leptin stimulates sympathetic nerve activity innervating white adipose tissue. Excitation of the sympathetic tone of white adipose tissue by chronic leptin administration resulted in a remarkable decrease in epididymal fat weight without variations in food intake, suggesting a role for leptin in the promotion of lipolysis [59]. However, there are some contradictory reports regarding the interaction of leptin and sympathetic nerve activity exerting a lipolytic action in white adipose tissue. A decrease in epidermal fat pads after leptin infusion was observed even in sympathectomized mice [60]. The autocrine and paracrine effects of leptin on adipocytes can explain the local effect on lipolysis [61]. A recent study revealed the involvement of the central nervous system in the lipolytic action of leptin; this activity was due to brain-derived neurotrophic factor (BDNF)-expressing neurons in the paraventricular nucleus [62]. Furthermore, leptin improves dyslipidemia via activation of 5′-adenosine-monophosphate-activated protein kinase (AMPK), which increases fatty acid oxidation in skeletal muscle [20]. In lipoprotein profiles, cholesterol levels in the LDL fraction were reduced by approximately 50% in leptin-treated mice compared with sham-treated control *ob/ob* mice [21]. In addition, the expression of LDL receptors and LDL uptake were significantly down-regulated by leptin treatment in HepG2 cells [63]. In vivo hyperleptinemia-induced rats had depleted TG content in liver, skeletal muscle, and pancreas without any increase in plasma FFA or ketones [64].

#### 3.1.2. Clinical Studies

Human studies using pedigree analysis demonstrated that leptin gene mutations and defects in the leptin receptor led to extreme hyperphagia and obesity [65]. Despite its anorexigenic effects, high leptin levels and leptin resistance caused by its low transport through the blood–brain barrier have been reported in obese patients [66]. Higher leptin levels were associated with an increased incidence of metabolic syndrome both in an observational study with a 14-year follow-up period [67] and in a study of schizophrenic patients [68]. In a 5-year prospective study of non-diabetic white men, higher leptin levels were associated with an increased risk of type 2 diabetes mellitus [69].

### 3.2. Adiponectin

Adiponectin is a 30 kDa protein that originates from adipose tissue. Adiponectin exists as multimers in plasma and has three major oligomeric forms combined with its collagen domain: a low-molecular-weight trimer, a middle-molecular-weight hexamer, and high-molecular-weight (HMW) 12- to 18-mers [70]. HMW adiponectin is a superior biomarker associated with protection against metabolic syndrome as the most potent form in the activation of AMP kinase [71].

#### 3.2.1. Experimental Studies

In animal experiments, the adiponectin level was reduced in obese animals, whereas caloric restriction led to adiponectin up-regulation [72]. The effect of adiponectin on adipose tissue differs depending on cell type. In mouse 3T3-L1 fibroblasts, adiponectin overexpression accelerated adipocyte differentiation and augmented lipid accumulation in fully-differentiated adipocytes [23]. Conversely, recombinant adiponectin blocked fat cell formation in bone marrow culture and inhibited the differentiation of cloned stromal pre-adipocytes [24]. The expression of both adiponectin receptors, AdipoR1 and AdipoR2, has been shown to decrease insulin resistance and obesity in mouse models [73] and is decreased in the muscle and adipose tissues of obese mice [74]. Adiponectin overexpression in adiponectin transgenic mice ameliorated insulin resistance and diabetes, while adiponectin-deficient mice showed a marked worsening of insulin resistance when administered a high-fat/high-sucrose diet [25]. The suggested mechanism of adiponectin’s effect on insulin sensitivity involves an increase in the phosphorylation and activation of AMPK in skeletal muscle and liver by adiponectin, leading to improved muscle fat oxidation and glucose transport [75]. Dyslipidemia was also found to be improved in adiponectin transgenic mice [76]. Adiponectin blocked the expression of macrophage scavenger receptor class A-1, which led to a reduction in the uptake of oxidized low-density lipoprotein (LDL) and inhibition of foam cell formation [26]. Elevated plasma adiponectin suppressed the development of atherosclerosis in apolipoprotein-E-deficient mice [77]. Furthermore, intimal smooth muscle cell proliferation was increased in adiponectin-deficient mice. However, this effect was reversed by adiponectin supplementation, suggesting that adiponectin plays an important role in remodeling blood vessels after endothelial injury [27].

#### 3.2.2. Clinical Studies

The plasma adiponectin level was reduced in obese patients, particularly those with visceral obesity [25]. The association of obesity and low adiponectin level suggested the possibility of adiponectin acting as a biomarker for metabolic syndrome, especially among perimenopausal and postmenopausal women or the elderly [78,79]. Low adiponectin levels were found in obese children with diabetes [80], and a polymorphism of the adiponectin gene resulting in a low adiponectin level was associated with diabetes [81]. Homeostatic model assessment (HOMA) including adiponectin level showed better prediction for metabolic syndrome than HOMA insulin resistance itself [82]. Low adiponectin levels were associated with a higher incidence of diabetes in prospective and longitudinal studies [83]. An increased level of adiponectin was associated with improvement of hepatic insulin resistance in severely obese women [84]. Thiazolidinedione, a therapeutic agent for type 2 diabetes, increases plasma adiponectin level [85]. The effect of thiazolidinedione on improved insulin resistance might depend in part on adiponectin.

### 3.3. Visfatin

Visfatin is a 52 kDa cytokine that functions like insulin and is expressed in various organs including skeletal muscle, liver, lymphocytes, and adipose tissue [86,87]. Visfatin was formerly known as NAMPT (or pre-B-colony-enhancing factor) and is a rate-limiting enzyme that converts nicotinamide to nicotinamide mononucleotide [86,87]. Visfatin is also released from visceral adipose tissue, predominantly from macrophages rather than from adipocytes [88].

#### 3.3.1. Experimental Studies

Obese diabetic mice were found to express more visfatin in visceral fat after weight gain [89]. Visfatin induces insulin resistance by regulating JAK2/STAT3 and IKK/NF-kB signaling [28]. Additionally, the administration of visfatin increases inflammatory cytokines such as interleukin (IL)-6, tumor necrosis factor-alpha (TNF-α), and IL-1β [90]. Visfatin induces endothelial dysfunction via the NF-kB pathway in the vascular endothelium [30]. Extracellular visfatin expression also promotes the proliferation of human vascular smooth muscle cells [31]. These processes associated with visfatin are implicated in accelerated atherosclerosis, in addition to the effects of visfatin on the accumulation of cholesterol in macrophages, both in vitro and in vivo [29,91].

#### 3.3.2. Clinical Studies

Among 350 obese women with metabolic syndrome, values of BMI, fat mass, and waist circumference were lower in patients in the highest tertile of visfatin level [92]. In another study, serum visfatin was significantly higher in obese children with metabolic syndrome compared with those having no components of metabolic syndrome. Visfatin level showed positive correlations with serum glucose, insulin, and HOMA-IR in obese subjects [93]. A meta-analysis pooling 13 observational studies demonstrated that plasma visfatin concentration was increased in participants diagnosed with overweight/obesity, type 2 diabetes mellitus, metabolic syndrome, or cardiovascular disease. This suggests a role for visfatin as a predictive biomarker for metabolic syndrome [94].

### 3.4. Fetuin-A

Fetuin-A, a 64 kDa glycoprotein also known as α2-Heremans–Schmid glycoprotein (AHSG), is mainly secreted from the liver and adipose tissue. Fetuin-A contributes to macrophage migration into adipose tissue [32] and augments the expression of proinflammatory cytokines such as IL-6 and TNF-α, while reducing adiponectin expression [95].

#### 3.4.1. Experimental Studies

A previous study reported that fetuin-A decreased insulin sensitivity by interfering with insulin receptor signaling at the tyrosine kinase level [96]. Fetuin-A-deficient mice showed improved insulin sensitivity and were resistant to diet-induced obesity [33]. However, the role of fetuin-A in cardiovascular events and complications associated with metabolic syndrome is controversial. Fetuin-A is implicated in atherogenesis by increasing macrophage foam cell formation by stimulating inflammatory cytokines and chemokines such as IL-6, monocyte chemotactic protein-1, intercellular adhesion molecule-1, and E-selectin expression in vitro [34].

#### 3.4.2. Clinical Studies

Regarding insulin resistance, higher fetuin-A levels were demonstrated to be associated with insulin resistance in polycystic ovarian syndrome and prepubertal children [97,98]. The fetuin-A level was positively associated with HOMA-IR and inversely associated with the quantitative insulin sensitivity check index. However, a protective role for fetuin-A against complications associated with metabolic syndrome, as mentioned above, has been demonstrated. Fetuin-A is an inhibitor of vascular calcification, especially among patients with chronic kidney disease (CKD) [99]. A low fetuin-A level was associated with greater aortic stiffness and vascular calcification in advanced CKD patients [35].

### 3.5. Plasminogen Activator Inhibitor-1 (PAI-1)

PAI-1, a physiological inhibitor of plasminogen activators and vitronectin, is synthesized in adipose tissue.

#### 3.5.1. Experimental Studies

Prior studies have revealed increased plasma PAI-1 levels with obesity and reduced levels with weight loss [100]. In obese rats, PAI-1 mRNA expression in visceral fat increased with the level of obesity [101]. Using cultured adipocytes from PAI-1+/+ and PAI-1-/- mice, the deletion of PAI-1 in adipocytes ameliorated insulin resistance by promoting glucose uptake and adipocyte differentiation [36]. Inhibition of plasma PAI-1 activity also resulted in improved hyperlipidemia in a diet-induced obese mouse model [37]. In mouse models, pharmacological inhibition of PAI-1 prevented hepatic steatosis and reduced serum cholesterol level by reducing PCSK9 synthesis [37]. PAI-1 also contributes to atherothrombosis. The deletion of PAI-1 inhibited carotid artery atherosclerosis [38], while pharmacological inhibition of PAI-1 attenuated atherosclerosis in an obese mouse model of metabolic syndrome by inhibiting macrophage accumulation and cell senescence in atherosclerotic plaques [102].

#### 3.5.2. Clinical Studies

The level of PAI-1 in the obese group was significantly higher than that in the control group, and PAI-1 was positively associated with components of metabolic syndrome such as higher BMI, skin fold thickness, blood pressure, LDL cholesterol level, and HOMA-IR [103].

### 3.6. Resistin

Resistin is part of a family of small, secreted cysteine-rich proteins with hormone-like activity that initiates inflammatory processes. In humans, peripheral blood mononuclear cells, macrophages, and bone marrow cells are the primary sources of circulating resistin; however, in rodents, resistin is primarily derived from adipose tissue.

#### 3.6.1. Experimental Studies

In obese mice, the plasma resistin level is significantly increased compared to healthy control mice [104]. Blockade of resistin improved blood glucose level in obese mice and increased glucose tolerance in healthy mice. In fact, resistin decreased glucose uptake in skeletal muscle cells independently of insulin-activated signaling pathways [39]. Resistin also exerts its glucoregulatory effects by stimulating hepatic gluconeogenesis [40]. Transgenic mice overexpressing human resistin showed features associated with decreased insulin sensitivity: accelerated inflammation in adipose tissue, increased lipolysis with increasing serum FFAs and glycerol, and accumulation of FFAs in skeletal muscle [41]. Resistin also contributed to foam cell formation by increasing lipid accumulation in macrophages. [42]. In vitro, resistin increased vascular cell adhesion molecule-1 and MCP-1 expression in endothelial cells [105] and promoted vascular smooth muscle cell apoptosis, which is linked to plaque vulnerability [43]. Elevated resistin levels were also detected in murine and human atherosclerotic lesions and in the serum of premature coronary artery disease patients, suggesting that resistin plays a role in cardiovascular disease and metabolic syndrome [106].

More recently, a potential mechanism for the pro-atherogenic role of resistin has been provided. This proposed mechanism suggests that resistin decreases the expression of LDL receptor (LDLR) in human hepatocytes in a PCSK9-dependent manner. Resistin, therefore, can affect serum lipid metabolism and cardiovascular disease by modulating PCSK9-induced LDLR expression [104].

#### 3.6.2. Clinical Studies

Elevated serum resistin levels were also detected in obese patients in an observational study [104]. In another cross-sectional study, a positive correlation between resistin level and clinical features such as waist circumference, systolic and diastolic blood pressure, plasma glucose, waist/hip ratio, serum triglyceride level, serum cholesterol level, serum VLDL level, plasma insulin level, and insulin resistance [107] was revealed.

### 3.7. Omentin-1

Omentin, also known as Intelectin-1, is a secretory glycoprotein that is highly and selectively expressed in visceral adipose tissue relative to subcutaneous adipose tissue. Both the omentin-1 gene and omentin-2 gene, a homolog which shares an 83% amino acid identity with omentin-1, are located in the chromosome 1q22-q23 region, which has been previously linked to type 2 diabetes [45].

#### 3.7.1. Experimental Studies

Omentin-1 increased insulin-stimulated glucose uptake in human adipocytes [108] and increased AMP-activated protein kinase (AMPK) phosphorylation in myocytes. This induced increased energy metabolism and fatty acid oxidation [45].

#### 3.7.2. Clinical Studies

Plasma levels of omentin-1 and omentin-1 gene expression were significantly higher in lean subjects than in obese and overweight subjects [109]. The plasma omentin-1 level inversely correlated with fasting glucose, BMI, waist circumference, and HOMA-IR [45]. Although there have been conflicting results on the level of omentin-1 in type 2 diabetic patients, lower omentin-1 levels were demonstrated in type 2 diabetes based on a meta-analysis of 42 clinical studies [44]. However, the omentin-1 level was higher in non-alcoholic fatty liver disease (NAFLD), which is known to be associated with insulin resistance and is a feature of metabolic syndrome [45]. Further studies are needed to identify the detailed mechanism in the association of omentin-1 with metabolic syndrome.

### 3.8. Lipocalin-2

Lipocalin-2 (LCN-2, also called neutrophil gelatinase-associated lipocalin) is a 25 kDa protein that plays a role in the innate immune response to bacterial infection. LCN-2 is expressed in various sites such as liver, kidney, brain, lung, and adipocytes [45].

#### 3.8.1. Experimental Studies

The level of expression of LCN-2 was higher in the adipose tissue and liver of diabetic/obese mice than in lean mice [48]. Regarding the role of LCN-2 in metabolic syndrome, increased lipid accumulation and insulin resistance were attenuated in LCN-2 knockout mice. Improved insulin sensitivity was demonstrated as a quicker decrease in glucose level after insulin injection in lipocalin-2 knockout mice than in wild-type mice [110]. LCN-2 was up-regulated in the livers of fatty liver Shionogi (FLS) mice, an animal model for non-alcoholic steatohepatitis (NASH) [47].

#### 3.8.2. Clinical Studies

The circulating LCN-2 level was positively correlated with adiposity (BMI, waist circumference, and body fat percentage), hypertriglyceridemia, hyperglycemia, and insulin resistance index (HOMA-IR). Furthermore, rosiglitazone treatment decreased the level of lipocalin-2 along with an increase in insulin sensitivity [48]. The LCN-2 level was higher in patients with metabolic syndrome than in controls [111].

### 3.9. Asprosin

Asprosin, encoded by two exons of the gene Fibrillin 1 (FBN1), is a novel adipokine discovered in a study of neonatal premature aging patients. Recent studies have found that asprosin is associated with metabolism and metabolic diseases including diabetes and obesity.

#### 3.9.1. Experimental Studies

In neonatal premature aging mice with an extremely lean body shape, decreased plasma asprosin level and decreased appetite were noted, suggesting an impact of asprosin in obesity by regulating appetite [50,112]. However, recombinant asprosin injection in mice did not change the body weight [113]. The hepatic asprosin level in type 1 diabetic mice was increased [114]. In addition, recombinant asprosin injection in mice induced hyperglycemia and hyperinsulinemia [50]. A recent study in type 1 diabetic mice induced by streptozotocin reported that aerobic exercise reduced liver asprosin level, to improve diabetes-related variables [114]. In addition, pretreatment with asprosin in mesenchymal stromal cells improved myocardial ejection function and reduced cardiac remodeling after myocardial infarction [115].

#### 3.9.2. Clinical Studies

The circulating asprosin level was positively associated with waist circumference and triglyceride level [50,51]. In a prospective cohort study, the fasting asprosin level was significantly higher in obese participants compared to normal weight participants [116]. In a cross-sectional study, higher plasma asprosin was found to be associated with impaired glucose regulation and type 2 diabetes [117]. In addition, an independent association between fasting glucose and serum asprosin in type 2 diabetes was demonstrated in a hospital-based case–control study [118].

### 3.10. Neuregulin 4

Neuregulin 4 (NRG4) is a novel brown-adipose-tissue-secreted adipokine with beneficial metabolic effects on obesity and its metabolic complications.

#### 3.10.1. Experimental Studies

*Nrg4*-deficient mice have increased insulin resistance and hepatic steatosis after high-fat diet consumption [52]. In contrast, *Nrg4*-overexpressing transgenic mice with high-fat diet exposure showed improved metabolic status compared to controls. Improved parameters included lower body weight gain and improved dyslipidemia and insulin sensitivity [52]. Furthermore, Nrg4 gene expression was decreased in white adipose tissue but not in brown adipose tissue of diet-induced obese mice [52,119,120].

#### 3.10.2. Clinical Studies

In a human study, the *Nrg4* mRNA level in adipose tissue was negatively associated with body fat mass and hepatic lipid content [52]. In addition, type 2 diabetes participants showed decreased expression of adipocyte NRG4 compared to that of participants with normal glucose control. In addition, NRG4 level was inversely associated with the severity of coronary artery disease (CAD) [121].

## 4. Molecular and Cellular Crosstalk in Central Obesity and Metabolic Syndrome

Obesity has generally been considered a risk factor for metabolic and cardiovascular diseases for decades. However, recently, the paradox of obesity has been highlighted from a new perspective in which the location of fat accumulation is the problem rather than the total amount of fat [122]. There are two types of adipose tissue, white and brown, which perform different functions [123]. In humans, white adipose tissue consists mainly of a central intra-abdominal component (visceral adipose tissue) associated with increased metabolic risk, whereas subcutaneous adipose tissue has a protective effect on energy homeostasis and cardiovascular health [124,125]. A previous study showed that human subcutaneous adipose tissue contains larger adipocytes, is less infiltrated by CD68+ and M1-activated cells, and expresses higher levels of cardioprotective adipokines such as adiponectin [126]. Additionally, intraperitoneal implantation of subcutaneous adipose tissue in obese mice prevented glucose intolerance and systemic inflammation [127]. However, the factors that determine visceral or subcutaneous fat distribution remain unknown.

Central obesity induces adipocyte hypertrophy and hyperplasia, macrophage infiltration, endothelial cell activation, and ectopic fat disposition due to excessive energy accumulation. Larger adipocytes are correlated with dysregulated adipokine expression, and hypertrophic adipocytes are prone to producing proinflammatory molecules [128]. In hypertrophic adipose tissue, local hypoxia can occur due to reduced blood flow relative to the size and number of adipocytes, which leads to reduced adiponectin production and increased proinflammatory cytokine expression [129]. Furthermore, obesity not only leads to increased macrophage infiltration in adipose tissue but also triggers their polarization as M1 macrophages producing proinflammatory cytokines and inducible nitric oxide synthase (iNOS) [130]. Through these processes, elevated cytokines and chemokines recruit monocytes that adhere to endothelial cells and elevate the expression of vascular adhesion molecules such as ICAM, VCAM, and E-selectin [131]. This ‘vicious cycle’ together with chronic inflammation in adipose tissue leads to various complications of metabolic syndrome such as hepatic fibroinflammatory injury, systemic arterial dysfunction, and insulin resistance. In this context, the adipokines and cytokines discussed above play pivotal roles in chronic inflammation, macrophage aggregation, and hypoxia and contribute to the variety of complications called metabolic syndrome (Figure 1).

## 5. The Roles and Associated Mechanisms of Adipokines in Cardiovascular Diseases

Clinical features of metabolic syndrome manifested by obesity, elevated blood pressure, and dyslipidemia have a significant impact on the increase in the risk of cardiovascular disease. In a previous study, cardiovascular and all-cause mortality were increased in patients with metabolic syndrome, independent of the baseline cardiovascular disease and diabetes status [132]. Excess weight gain causes an increase in angiotensin II and aldosterone, which regulate renal sodium excretion and play a critical role in blood pressure regulation [133,134]. As an endocrine organ, adipose tissue synthesizes and releases peptides and nonpeptide compounds that have a role in cardiovascular homeostasis. Obesity and enlarged adipose tissue enhance the production of metabolic products that are widely related to atherosclerosis, endothelial dysfunction, hypertension, and dyslipidemia.

### 5.1. Adiponectin

Several animal studies reported multiple salutary effects of adiponectin on cardiovascular health. Adiponectin knockout mice showed progressively increased leukocyte–endothelium interactions in microcirculation, and treatment with the globular adiponectin domain in adiponectin knockout mice normalized leukocyte rolling flux and leukocyte adhesion to values similar to those in control WT mice. In addition, adiponectin knockout mice exhibited a significantly increased expression of E-selectin, which is implicated in leukocyte rolling and leukocyte adhesion [135]. Aortic ring tissues derived from adiponectin knockout mice showed a decrease in endothelial NOS expression that might cause a defect in vasodilation. This was reversed by treatment with recombinant adiponectin [136]. In another study, adiponectin administration in obese rats increased endothelial NOS by activating the AMPK pathway and promoting NO production. This resulted in the relaxation of the aortic ring [137]. Decreased vasodilatation of the coronary artery and aorta in response to acetylcholine was demonstrated in *db/db* mice, and the response was recovered after adiponectin administration. In cultured human umbilical vein endothelial cells, adiponectin showed a protective effect against angiotensin-II-induced vascular endothelial damage [138]. In addition, adiponectin attenuated angiotensin-II-induced NADPH oxidase activation in renal proximal tubular cells [139]. These studies suggest that adiponectin production is closely related to endothelial function in vasodilation.

In addition to its effect on the vasculature, adiponectin has a direct effect on cardiomyocytes. Significant concentric cardiac hypertrophy was induced by pressure overload in adiponectin-deficient mice [140]. Increased myocardial infarction size and further deterioration of cardiac function after ischemia–reperfusion injury (IRI) were observed in adiponectin-deficient mice compared to controls [141]. A single injection of recombinant adiponectin through the coronary artery during ischemia also led to a reduction of myocardial infarction size and improvement in left ventricular function in the myocardial IRI of pigs [142]. Adiponectin administration improved the contractility of cardiomyocytes in *db/db* obese mice, and the alleviation of endoplasmic reticulum stress mediated by adiponectin might be associated with this effect [143]. The anti-apoptotic activity of adiponectin has been suggested to mitigate IR injury in cardiomyocytes. The adiponectin-mediated regulation of cyclooxygenase-2 (COX-2) expression via a sphingosine kinase-1 (SphK-1)- and sphingosine-1-phosphate (S1P)-dependent mechanism has also been suggested as one of the mechanisms of myocardial protection by adiponectin after IRI. Adiponectin was required to fully induce the cardioprotective effect of COX-2, and treatment with SphK-1 inhibition or an S1P antagonist reduced COX-2 production stimulated by adiponectin. Another study demonstrated the association of the adiponectin–AMPK–endothelial NO synthesis pathway in myocardial IRI. Adiponectin administration significantly improved left ventricular function and coronary flow after IRI and limited the size of myocardial infarction along with AMPK activation. Inhibition of NO synthase abrogated adiponectin protection from myocardial contractile dysfunction after IRI [144].

### 5.2. Leptin

The role of leptin in the cardiovascular system remains controversial. Leptin administration for 12 days increased the arterial pressure and heart rate in rats [145]. Further, humans with loss-of-function mutations in leptin showed low BP despite severe obesity [146]. However, leptin had concentration-dependent negative inotropic effects on myocyte contraction and intracellular Ca^2+^ release in rat ventricular myocytes [147]. Mice with cardiomyocyte-specific deletion of the leptin receptor (LEPr) showed decreased ejection fraction of the heart due to impairment of energy production through AMPK and mTOR signaling. Cardiomyocyte-specific LEPr-/- mice showed more severe cardiac dysfunction after myocardial infarction [148]. Although leptin receptors are found on endothelial cells [149], the vasoregulatory effects of leptin are uncertain. Two studies showed that leptin increased the vasodilatation of rat aortic rings in vitro via a nitric oxide (NO)-dependent mechanism [150], but another study showed that leptin had no effect on hemodynamics, even after blocking NO generation [151]. Moreover, leptin synthesis was found to increase when cultured with angiotensin II adipose cells and rats in vivo [152]. Leptin with adipose-tissue-derived angiotensin II can promote obesity-related hypertension [153]. In an in vitro study using endothelial cells of the human umbilical vein, leptin induced chronic oxidative stress in endothelial cells and promoted atherogenesis [154]. Leptin administration also stimulated the proliferation of rat vascular smooth muscle cells and regulated osteoblastic differentiation and calcification of vascular cells. When vascular cells underwent osteoblastic differentiation producing calcified particles, leptin treatment augmented the level of calcification and alkaline phosphatase activity, a marker of osteogenic differentiation of osteoblastic cells, in a time-dependent manner after leptin exposure. Moreover, the leptin receptor was found in calcifying vascular cells [155].

In studies based on clinical data, there have been controversial reports about the role of leptin in cardiovascular disease. Leptin was a predictor of myocardial infarction, coronary events, and stroke independent of body mass index (BMI) [156]. Further, leptin levels were significantly higher in men who experienced a coronary event than in others without such events [157]. In a prospective study of 4080 men, increased BMI and leptin levels were demonstrated as independent predictors for the incidence of heart failure [158]. In the evaluation of 818 elderly Framingham Study participants, leptin level was strongly associated with the incidence of congestive heart failure and cardiovascular disease [159]. However, the Multi-Ethnic Study of Atherosclerosis, with 1905 participants without underlying cardiovascular disease, showed a different result, i.e., that leptin level was not associated with the incidence of cardiovascular events after adjustment for various cardiovascular risk factors including BMI [160].

### 5.3. PAI-1

PAI-1, a proinflammatory adipokine, is increased in metabolic syndrome and obesity. PAI-1 has established roles in different pathways for atherosclerosis and cardiovascular risk. An increase in PAI-1 down-regulates tissue plasminogen activator and urokinase activity. This triggers a prothrombotic state and contributes to the development of cardiovascular events. Transient venous thrombosis in the tail and hind limbs was demonstrated in transgenic mice overexpressing PAI-1 [161]. When transgenic mice were generated to express human PA1-1, elevated levels led to spontaneous multiple coronary arterial occlusions and subendocardial infarction in 90% of transgenic mice older than 6 months [162].

### 5.4. Omentin-1

Omentin-1 significantly suppressed foam cell formation in human macrophages. Chronic infusion of omentin-1 into apoE knockout mice reduced atherosclerotic lesions and decreased infiltration of macrophages into atherosclerotic plaques [46]. In a cross-sectional study of type 2 DM patients, omentin-1 was inversely related to arterial stiffness and carotid plaque formation. However, plasma omentin-1 level was significantly elevated in patients with CAD compared to non-CAD patients or healthy volunteers [45]. A recent prospective, 14-year follow-up study of diabetic patients without a previous cardiovascular event observed that higher omentin-1 levels were associated with a higher risk of cardiovascular events, composite incidence of myocardial infarction, stroke, or cardiovascular death, even after adjustment for other risk factors [163].

### 5.5. Lipocalin-2

Cardiac stress induces an increase in LCN-2 expression. Elevated LCN-2 levels were observed in cardiomyocytes of a rat model of post-MI heart failure. High levels of LCN-2 were also detected in the intima of cardiac vasculature after hypoxic stress [110]. LCN-2 expression was increased in atherosclerotic plaques of apolipoprotein E knockout mice compared with control mice [49]. Elevated local levels of lipocalin-2 can mediate cardiovascular function by promoting inflammatory cytokines such as MMP9 and apoptosis [110].

The challenge of modulating and interpreting the dynamic function of adipokines in cardiovascular organs could provide a remarkable opportunity to alter the course of cardiovascular complications in metabolic syndrome.

## 6. Future Perspective

Adipocytes have endocrine functions with a wide range of effects on other organs by releasing adipokines that have important roles in the regulation of the energy balance and glucose homeostasis. This provides a close association with the incidence of metabolic syndrome. Adipokines are also implicated in cardiovascular disease, which is the most serious complication of metabolic syndrome. Given these roles, the possibility of utilizing adipokines as diagnostic biomarkers or therapeutic targets for metabolic syndrome has been highlighted. Currently, at least 615 adipokines have been identified as part of the adipocyte secretome [164]. With the advancement of proteomics and metabolomics analysis tools, multiple studies have been conducted to identify novel adipokines related to metabolic syndrome, in addition to traditional adipokines. One study revealed 22 cardiometabolic proteins related to metabolic syndrome using targeted proteomics to assess 249 proteins among 2444 participants [165]. Another case–control study used untargeted metabolomics with clinical, sociodemographic, and food habit parameters to describe early phenotypes and build multidimensional predictive models for metabolic syndrome [166]. Using a highly multiplexed, aptamer-based, affinity proteomics platform, 1095 plasma proteins were assessed for association with metabolic syndrome [167]. Future studies using various multi-omics platforms including proteomics and metabolomics could provide an insight into the molecular pathogenic pathway of metabolic syndrome and help to identify new adipokines as candidates for predictive biomarkers or therapeutic targets of metabolic syndrome.

In addition to the efforts to discover novel adipokines as biomarkers, well-designed experimental and clinical studies are required to validate their role. The action of adipokines is complex and has mainly been demonstrated based on data obtained in vitro and in some animal models. Further studies, especially human studies examining the role of specific adipokines and their interactions with metabolic complications, are needed. From these studies, major pathogenic adipokines associated with metabolic syndrome can be specified, and therapeutic approaches to modify the expression and course of metabolic syndrome and its associated mechanisms can be followed. The favorable consequences for metabolic syndrome of several anti-diabetic medications such as metformin and thiazolidinedione, along with modulation of the levels of adipokines, have been demonstrated [48,85,168]. Further therapeutic options focusing particularly on the selective beneficial effects of adipokines should be developed in the future.

The contributions of individual adipokines to the pathophysiological features of metabolic syndrome remain controversial. Future studies should provide new paradigms regarding the undiscovered roles of various adipokines, and these might unveil new treatments for metabolic syndrome based on a better understanding of its etiology and associated pathophysiology.

## Figures and Tables

**Figure 1 molecules-27-00334-f001:**
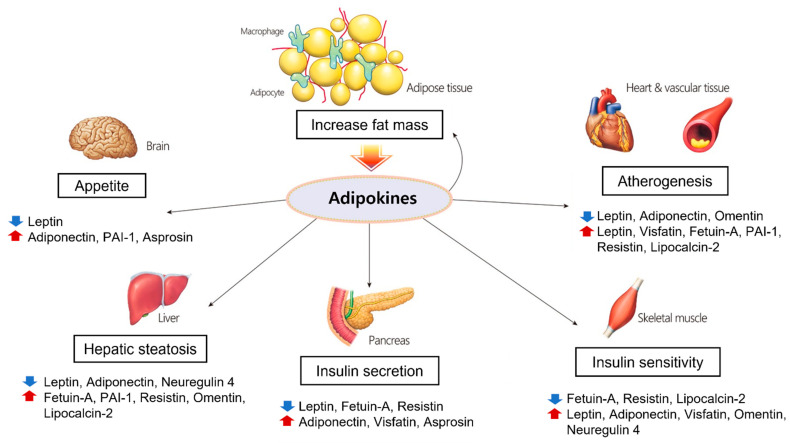
Physiological processes regulated by adipokines in each organ. Adipokines secreted from adipose tissue play important roles in adiposity, glucose and lipid metabolism, and atherosclerosis.

**Table 1 molecules-27-00334-t001:** Summary of the effect of each adipokine on development of metabolic syndrome.

Adipokines	Role in Development of Metabolic Syndrome
Leptin	Increase [16] or decrease [17] in adipogenesis dependent on dosage Inhibition of insulin secretion [18], improvement of insulin sensitivity [19] Increase in fatty acid oxidation [20], decrease in circulating triglyceride level [21] Increase in blood pressure by sympathetic activation [22]
Adiponectin	Increase [23] or decrease [24] in adipocyte differentiation Improvement in insulin sensitivity [25] Decrease in foam cell formation [26], inhibition of intimal smooth muscle cell proliferation [27]
Visfatin	Decrease in insulin sensitivity [28] Increase in accumulation of cholesterol in macrophages [29], inducement of endothelial dysfunction [30], promotion of proliferation of vascular smooth muscle cells [31]
Fetuin-A	Increase in macrophage infiltration into adipose tissue [32] Decrease in insulin sensitivity [33] Increase in foam cell formation [34], prevention of arterial calcification [35]
PAI-1	Decrease in insulin sensitivity [36] Increase in liver steatosis, increase in serum cholesterol [37] Increase in atherogenesis with macrophage accumulation [38]
Resistin	Decrease in insulin sensitivity [39], increase in hepatic gluconeogenesis [40] Increase in lipolysis with increased serum free fatty acid and glycerol [41] Increase in foam cell formation [42], promotion of vascular smooth muscle cell apoptosis [43]
Omentin-1	Improvement in insulin sensitivity [44] Improvement in energy metabolism [45] Decrease in foam cell formation, size of atherosclerotic plaque, and infiltration of macrophages [46]
Lipocalin-2	Increase in insulin resistance [47] Increase in lipid accumulation [48] Increase in atherogenesis [49]
Asprosin	Increase in insulin secretion [50] Increase in serum triglyceride level [50,51]
Neuregulin 4	Improvement in insulin sensitivity [52] Decrease in liver steatosis [52]

Abbreviations: PAI-1, Plasminogen activator inhibitor-1.

## Data Availability

All data is provided in full in the body of this paper.

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
