# Peer review of "The Roles and Associated Mechanisms of Adipokines in Development of Metabolic Syndrome"

_molecules, 2022, doi:10.3390/molecules27020334_

Round 1

Reviewer 1 Report

Blind Comments to Author

The manuscript ” The role of adipokines and inflammatory cytokines in the development of metabolic syndrome: a comprehensive review “ by Ji Eun Kim , Jin Sun Kim and coauthors summarize the molecular mediators thought to be associated with metabolic syndrome, which may serve as potential therapeutic targets to prevent or treat this syndrome.

The review deals with a very important topic and high clinical impact in the field. As stated in section 6 by the authors, newly identified adipokines and adipokine patterns using omics techniques underline the need to highlight the roles of certain metabolic syndrome associated mediators. However, in my opinion the presented manuscript fails to sum up new findings and discuss them in the context of potential therapeutic targets. Even if the manuscripts sums up nearly 300 references, the majority is older than 5 years. Beside the classic adipokines only visfatin, Fetuin-A and PAI were highlighted. It is unclear how this list of adipokines was concluded when already a good review about the same topic published in molecules in 2020 (doi:10.3390/molecules25215218) sums up a more differentiated list of adipokines. Therefore, the presented information are in majority not up to date and not comprehensive as predicted by the title.

In contrast to the above mentioned Review from 2020, the presented work focus more on in vivo and in vitro studies as stated in line 68,69. Both kind of studies are essential tools in research. However when the goal is to discuss the findings in a clinical and pharmaceutical context as stated in line 39-41 I expect to discuss these kind of studies in comparison to the human situation. This was not sufficiently performed. Consequently, the review is only a summery of studies and does not offer new findings, conclusions and recommendations.

Major concerns are:

  • It is not clear how the list of highlighted adipokines was concluded or what it based of
  • To many and to old references
  • Using the term recent or previous in relation with references older than 5 years
  • Missing discussion of in vivo findings in relation to the clinical or human situation
  • The final recommendations are to general and are a missed chance to highlight key studies or from the in vivo or in vitro research
  • Figure 1 /Table 2 provides information about brain and pancreas, two tissues which were referenced 2 times and 1 time, respectively in the Leptin paragraph. Therefore, the figure/ table does not reflect the information/content of the text.

Minor concerns are:

  • Inflammatory cytokines were introduced by the title but later summed up under adipokines
  • The structure of single paragraphs jumps between in vitro, in vivo and clinical study, is therefore confusing and could be improved

Because the title and the goal was not sufficiently elaborated and major conclusions of in vitro and in vivo studies in the human context were missing the review fails to present new information. The Review about Adipokines published in molecules in 2020 showed newer and more relevant information. Therefore, I recommend rejection of the presented manuscript.

Author Response

Referee 1.

  1. However, in my opinion the presented manuscript fails to sum up new findings and discuss them in the context of potential therapeutic targets. Even if the manuscripts sums up nearly 300 references, the majority is older than 5 years. Beside the classic adipokines only visfatin, Fetuin-A and PAI were highlighted. It is unclear how this list of adipokines was concluded when already a good review about the same topic published in molecules in 2020 (doi:10.3390/molecules25215218) sums up a more differentiated list of adipokines. Therefore, the presented information are in majority not up to date and not comprehensive as predicted by the title.
  • In the original manuscript what we first submitted, we focused to summarize the associated mechanism of adipokines and cytokines in metabolic syndrome. For the purpose, the most extensively studied adipokines and cytokines, which may be belonged to classic ones were chosen at first. So, we agree that our work was not sufficient to sum up recent adipokines, as reviewer’s comment. We deeply appreciate reviewer’s precise comment and tried to revise the manuscript including recent researches and remove old references. The description about adipokines in recent researches associated with metabolic syndrome was added line 277-357 in revised manuscript.
  1. In contrast to the above mentioned Review from 2020, the presented work focus more on in vivo and in vitro studies as stated in line 68,69. Both kind of studies are essential tools in research. However when the goal is to discuss the findings in a clinical and pharmaceutical context as stated in line 39-41 I expect to discuss these kind of studies in comparison to the human situation. This was not sufficiently performed. Consequently, the review is only a summery of studies and does not offer new findings, conclusions and recommendations.
  • We appreciate reviewer’s precise comment. Our previous manuscript focused to describe both the action of adipokines itself and associated mechanisms. For the reason, many in vivo and in vitro studies were included. However, we totally agree with reviewer that it is essential to interpret those results based on clinical studies and validate those findings with human studies, and our previous manuscript might not be sufficient to verify the results from animal studies in human studies. So we tried to clarify whether the study results come from animal studies or human studies and describe the results separately for each adipokine. We believe it helps to identify the findings from animal studies are consistently found in human studies, which confirm the role of each adipokine in metabolic syndrome.

Major concerns are:

  1. It is not clear how the list of highlighted adipokines was concluded or what it based of
  • As we mentioned above, we chose the list of adipokines which have been studied the most extensively about the mechanisms of adipokines in the occurrence of metabolic syndrome. However, we agree with reviewer that recent researches should be included, and tried to describe newly investigated adipokines studied both in animal and human studies.
  1. To many and to old references, using the term recent or previous in relation with references older than 5 years
  • We appreciate reviewer’s precise comment. We tried to remove old references published more than 5 years ago as much as possible. Only valuable references containing core results which should be emphasized and not be replaced with other recent researches were remained.
  1. Missing discussion of in vivo findings in relation to the clinical or human situation
  • We appreciate reviewer’s precise comment. For this matter, we reorganized the results separately from experimental and clinical studies, and tried to show that the findings from animal studies were consistently found in human studies, which help to understand the role of adipokines in metabolic syndrome
  1. The final recommendations are to general and are a missed chance to highlight key studies or from the in vivo or in vitro research
  • We appreciate reviewer’s precise comment. The main purpose of researches about the role of adipokines is believed to reveal the possibility as the diagnostic biomarkers and new therapeutic targets. Although utilization of adipokines for the purpose has not been established yet, we tried to summarize the value of adipokines associated with the features of metabolic syndrome, which may also be link to the prognostic value. In the future perspective, we focused that we may have higher chance of revealing more useful adipokines according to the development of diagnostic technique. And therapeutic approaches associated with modulating the levels of adipokines were mentioned, though there have not been studied much. We tried to revise them to emphasize and highlight the messages.
  1. Figure 1 /Table 2 provides information about brain and pancreas, two tissues which were referenced 2 times and 1 time, respectively in the Leptin paragraph. Therefore, the figure/ table does not reflect the information/content of the text.
  • As reviewer mentioned, we revised the figure 1 to represent the action of each adipokine in target organ and tried to put all the data we mentioned in the content.

Minor concerns are:

  1. Inflammatory cytokines were introduced by the title but later summed up under adipokines
  • As reviewer mentioned, we revised the manuscript focusing adipokines rather than describing cytokines together. The parts of cytokines were removed.
  1. The structure of single paragraphs jumps between in vitro, in vivo and clinical study, is therefore confusing and could be improved
  • We tried to clarify whether the study results come from animal studies or human studies. For the purpose the results from in vitro, in vivo and clinical study were described separately.
  1. Because the title and the goal was not sufficiently elaborated and major conclusions of in vitro and in vivo studies in the human context were missing the review fails to present new information. The Review about Adipokines published in molecules in 2020 showed newer and more relevant information. Therefore, I recommend rejection of the presented manuscript.
  • According to reviewer’s comment, we tried to revise the manuscript intensively including new information. Although valuable information about the association of adipokines with cardiometabolic diseases was shown in previous review in 2020, our manuscript tried to show the detailed mechanisms regarding the association of each adipokine with metabolic syndrome, which may help to understand the role of adipokines and elucidate therapeutic targets. We hope reviewer to see the possibility of our manuscript.

Reviewer 2 Report

The manuscript by Kim and colleagues have comprehensively summarized the contribution of adipokines and cytokines in the metabolic syndrome development, focusing on both glucose and lipid metabolism.

Authors might need to decide whether they want to use "adipocytokines" or "adipokines" in this manuscript. 

Lines 64-69 look weird in this manuscript. is there any reason putting this information here in the section 2? Please explain. 

Author Response

Referee 2.

  1. Authors might need to decide whether they want to use "adipocytokines" or "adipokines" in this manuscript. 
  • We appreciate reviewer’s precise comment. As reviewer mentioned, both terms should be clearly described if they should be used both. For clarifying our messages, cytokines from adipose tissues were removed from the context and only adipokines from adipocytes were summarized. So, the term,”adipokines” was only used.
  1. Lines 64-69 look weird in this manuscript. is there any reason putting this information here in the section 2? Please explain. 
  • In our previous submission of this manuscript, one of reviewers pointed out we should mention precisely about the methods how we searched and gathered the related materials. If it looks weird, the part could be removed.

Reviewer 3 Report

In this manuscript, the authors reviewed the associations of adipocytokines and the metabolic syndrome as well as its associated complications. The topic is of interest, the literature review is quite comprehensive, and the English presentation is readable throughout the manuscript. The perspectives for future research are also suggested. However, there are some points of that the authors may consider to improve the data presentation.

  1. It is not clear what the main message of the Fig 1 is giving. There is no description in the figure legend for the interpretation of the symbols (e.g. the up and down arrows) and no information for the diagram which makes this figure unable to stand alone. As in the main text of the article explains the molecular mechanisms of each adipocytokine on the development of metabolic syndrome and its associated complications, the figure should also support the main text by schematically summarizing the mechanisms that may promote or prevent metabolic syndrome. The data in Table 1 may be transformed to the figure in which it will be more attractive and easy to get understand.
  2. The term "adipokine" or "adipocytokine" should be used consistently or synchronously for the whole manuscript. The reader can understand that adipokines are produced by adipocytes whereas adipocytokines are produced by other cells in adipose tissue which induced by obesity or hyper-adiposity. However, the pattern of using these term in this manuscript is not clear. If the terms are interchangeable in this article, it should be stated in the introduction. On the other hand, if they are not interchangeable, the definition of each term should be also given. 
  3. Related to (2.) the table summarizes the source of each adipokine or adipocytokine mentioned in this article should be given since adipose tissue is a big ecosystem consisted of various cell types.
  4. The abbreviations in the abstract should be spelled out.
  5. Please check (or correct) the Lines 324-326, the reference of GLUT4 as a receptor of insulin should be added.
  6. There are some typos (for example al-dosterone) needed to be corrected.

Author Response

Referee 3.

  1. It is not clear what the main message of the Fig 1 is giving. There is no description in the figure legend for the interpretation of the symbols (e.g. the up and down arrows) and no information for the diagram which makes this figure unable to stand alone. As in the main text of the article explains the molecular mechanisms of each adipocytokine on the development of metabolic syndrome and its associated complications, the figure should also support the main text by schematically summarizing the mechanisms that may promote or prevent metabolic syndrome. The data in Table 1 may be transformed to the figure in which it will be more attractive and easy to get understand.
  • We appreciate reviewer’s precise comment. As reviewer mentioned, the concept of figure 1 was not clear. We tried to summarize the action of each cytokine in the target organs at first, and revised the figure 1 more clearly as reviewer mentioned.
  1. The term "adipokine" or "adipocytokine" should be used consistently or synchronously for the whole manuscript. The reader can understand that adipokines are produced by adipocytes whereas adipocytokines are produced by other cells in adipose tissue which induced by obesity or hyper-adiposity. However, the pattern of using these term in this manuscript is not clear. If the terms are interchangeable in this article, it should be stated in the introduction. On the other hand, if they are not interchangeable, the definition of each term should be also given. 
  • We appreciate reviewer’s precise comment. As reviewer mentioned, both terms should be clearly described if they should be used both. For clarifying our messages, cytokines from other cells in adipose tissues were removed from the context and only adipokines from adipocytes were summarized. So, the term,”adipokines” was only used in the revised manuscript.
  1. Related to (2.) the table summarizes the source of each adipokine or adipocytokine mentioned in this article should be given since adipose tissue is a big ecosystem consisted of various cell types.
  • We appreciate reviewer’s precise comment. We focused on only adipokines from adipocytes tried to clarify the target organ of each adipokine in figure 1 merging with the content of table 2.
  1. The abbreviations in the abstract should be spelled out.
  • We appreciate reviewer’s precise comment. It was spelled out.
  1. Please check (or correct) the Lines 324-326, the reference of GLUT4 as a receptor of insulin should be added.
  • We appreciate reviewer’s precise comment. In our previous manuscript, insulin resistance was described as a complication of metabolic syndrome, and detailed mechanisms of insulin resistance in metabolic syndrome were described. When we revised the manuscript, we thought it should be described as one of features of metabolic syndrome to reveal the role of each adipokine in insulin resistance. So the precise mechanisms of insulin resistance itself were removed.
  1. There are some typos (for example al-dosterone) needed to be corrected.
  • We appreciate reviewer’s precise comment. It was corrected including other mistakes.

Round 2

Reviewer 1 Report

The manuscript was revised extensively and improved a lot. I appreciate that the literature was reduced and updated. Also the differentiation between experimental and clinical studies adds more structure and adds value as it becomes clear how the experimental findings should be judged. By cross reading I noticed some language mistakes. Therefore, I recommend minor English language check before publication. After minor language editing I would recommend to accept the manuscript for publication.

Author Response

Reviewer 1.

I recommend minor English language check before publication. After minor language editing I would recommend to accept the manuscript for publication.

  • We deeply appreciate the thorough review. Actually we received language editing service twice before the resubmission. However, some errors may have been remained. This time we asked the service to different company. Corrections were typed in re-colored text. Hope the results look okay for publication.

Reviewer 3 Report

All concerns raised by this reviewer are adequately addressed. This manuscript may be accepted in the present form.

Author Response

Deeply appreciate your generous decision and comments.